# R-Bench: Are your Large Multimodal Model Robust to Real-world Corruptions?

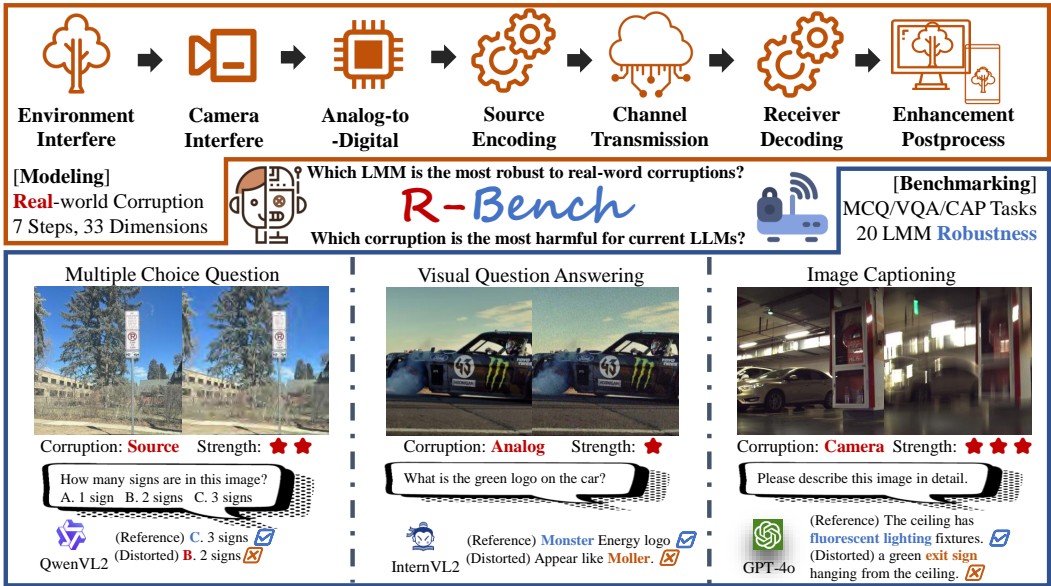

Figure 1: The construction overview of the proposed **R-Bench**. We model the full-link real-world corruption and evaluate the performance of LMMs on reference/distorted (left/right) images. Experiments demonstrate that the LMMs **solve the original image** but **hallucinate against corruption**.

## Abstract

The outstanding performance of Large Multimodal Models (LMMs) has made them widely applied in vision-related tasks. However, various corruptions in the real world mean that images will not be as ideal as in simulations, presenting significant challenges for the practical application of LMMs. To address this issue, we introduce R-Bench, a benchmark focused on the **Real-world Robustness of LMMs**. Specifically, we: (a) model the complete link from user capture to LMMs reception, comprising 33 corruption dimensions, including 7 steps according to the corruption sequence, and 7 groups based on low-level attributes; (b) collect reference/distorted image dataset before/after corruption, including 2,970 question-answer pairs with human labeling; (c) propose comprehensive evaluation for absolute/relative robustness and benchmark 20 mainstream LMMs. Results show that while LMMs can correctly handle the original reference images, their performance is not stable when faced with distorted images, and there is a significant gap in robustness compared to the human visual system. We hope that R-Bench will inspire improving the robustness of LMMs, **extending them from experimental simulations to the real-world application**.

## 1 Introduction

Large Multimodal Models (LMMs) have demonstrated outstanding abilities in a wide range of visual tasks. Due to the cross-modal interaction capabilities brought by instruction tuning, they can

accurately understand visual information and provide precise feedback based on queries. However, unlike single-modal Large Language Models (LLMs), which have become the foundation of daily life for humans, satisfying diverse needs such as writing, searching, and coding, LMMs, despite their excellent performance, have not yet reached the same status. From the neural perception perspective (Zhang et al., 2022), the visual cortex accounts for at least 70% of the external information processing; and according to Cisco statistics (Cisco, 2020) from 2018-2023, image/video data accounted for 82% of network bandwidth. Therefore, if the real-world applications of LMMs can be expanded from text to images, it will bring tenfold convenience to daily human life.

**Why are LMMs excellent in benchmarks but limited in the real-world?** Robustness is a crucial factor. In experiments, LMMs usually receive high-quality images, but in real-world scenarios that includes numerous corruption, such as object motion, lens blur, etc. Worse still, in embodied AI or mobile devices (Bai et al., 2024) where agents call LMMs to perform tasks, due to the limitations of edge computing power, current models are mainly deployed on the cloud. The complex transmission process is also risky for corruption. Considering the image modality is much larger than text and encounters more losses in the real-world, LMMs must ensure robust results on distorted content.

Unfortunately, despite the emergence of benchmarks for LMMs in numerous tasks over the past few years, assessing their robustness in the real-world remains an unexplored challenge. First, the evaluation of robustness requires images before and after distortion, which presents a significant challenge in data collection. This involves modeling and categorizing corruptions on one hand, and maintaining carefully curated reference/distorted image pairs (rather than taking a bunch of distorted images directly) on the other. In addition, unlike the commonly used benchmark dimensions such as accuracy/recall, robustness currently lacks a universally accepted metric.

Considering these issues, we have established R-Bench to evaluate the robustness of LMMs in the real world. R-Bench aims to test the resistance of different LMMs to corruptions and to identify the most significant corruptions affecting LMMs' performance, thereby pointing out optimization directions for future LMMs and helping them adapt to real-world images, as shown in Figure 1. Our contributions can be summarized as follows:

- A comprehensive modeling of corruption to date. We have considered the entire link from image capturing to LMMs finally seeing the image based on knowledge in imaging science and communication engineering, and categorized it into 7 steps, and by underlying features into 7 groups, totaling 33 corruption dimensions in 3 different strengths.

- A large dataset containing 2,970 pairs of corresponding reference/distorted images, meticulously annotated by human experts, covering the three most mainstream tasks of LMMs. The data characteristics prove that they are suitable as a testing sequence for robustness.

- A comprehensive benchmark experiment that considers the performance of 20 mainstream LMMs on reference/distorted images, thereby measuring robustness. In particular, we have proposed the concepts of absolute/relative robustness with a mathematical definition, establishing a standardized process for future robustness evaluation.

## 2 RELATED WORKS

For the task of generating text from images, its robustness has been the long-term research topic as listed in Table 1. However, a series of works, including RobustBench, MSRVTT-P, AttackVLM, and OpenRedTeaming (Croce et al., 2021; Schiappa et al., 2022; Zhao et al., 2023; Cui et al., 2024), have treated robustness as a typical adversarial task. The distortions in images come from manual attacks, such as manually adjusting a part of the image or injecting carefully designed noise to induce the model to make mistakes. Although they study robustness, these distortions are completely different from the corruption in the real world. RobustVLM (Schlarmann & Hein, 2023) is the first study on corruption robustness, but its range of distortions is not rich enough. Subsequently, MMRobustness (Qiu et al., 2024) and MMC-Bench (Zhang et al., 2024a) considered more detailed distortion scenarios. However, their assessments of robustness are not reasonable enough; the former directly uses the task score of the distorted images as robustness, while the latter measures the similarity between the distorted images and the answers to the original images. Both of these evaluation mechanisms are irrational, which will be analyzed in Section 3.3.

Table 1: Comparision between previous robustness-related benchmark and R-Bench. R-Bench is more comprehensive and reliable in real-world evaluations.

| Benchmark | Mechanism | Dimensions | Task | Robustness |
|---|---|---|---|---|
| RobustBench | Handcraft Attack | 15 | MCQ | Absolute |
| MSRVTT-P | Handcraft Attack | 7 | CAP | Absolute, Similarity |
| RobustVLM | Machine Corruption | 2 | CAP | Absolute, Similarity |
| AttackVLM | Generative Attack | 2 | CAP | Similarity |
| OpenRedTeaming | Handcraft Attack | 3 | MCQ, VQA, CAP | Absolute |
| MMRobustness | Machine Corruption | 14 | VQA, CAP | Absolute |
| MMC-Bench | Machine Corruption | 29 | CAP | Similarity |
| **R-Bench** | **In-the-wild, Machine Corruption** | **33** | **MCQ, VQA, CAP** | **Absolute, Relative** |

In addition, all previous research on robustness has two common issues. First, very few works can simultaneously consider the three classic LMM tasks, namely Multiple Choice Questions (MCQ), Visual Question Answering (VQA), and Captioning (CAP). This limits the credibility of the benchmark in terms of robustness. Moreover, although some corruption comes from the real-world, they only include the machine transmission process from the captured image to LMM reception; they ignore the former process in obtaining the image itself, namely in-the-wild corruption. Therefore, an objective and reliable benchmark needs to be conducted to analyze robustness across multiple tasks and dimensions along the entire real-world corruption link.

## 3 BENCHMARK CONSTRUCTION

### 3.1 REFERENCE DATA COLLECTION

To comprehensively characterize image data from the real world, we collect high-quality reference data and then add corruption to obtain distorted images. The selection of references is based on three principles: (1) Diversity: The data must contain different subjects, backgrounds, styles, etc., and the three tasks should be as evenly distributed as possible to avoid affecting the credibility of the benchmark due to highly consistent data. (2) Reality: The images must come from natural scenes, such as UGC (user-generated content) taken by average users. Content commonly found in other benchmarks, such as anime (Li et al., 2022), screen content (Li et al., 2024d), and AI-Generated Content (AIGC) (Li et al., 2024b;c; 2023) will be filtered out. (3) Quality: As high-quality reference information, the images must not already be distorted, as otherwise, they cannot be distinguished from the corresponding distorted images.

To obtain reference images, we have implemented the following mechanisms: First, we considered samples from today's mainstream benchmarks, including seven LMM benchmark datasets for MCQ, VQA, and CAP tasks. (Lin et al., 2014; Liu et al., 2023c; Wu et al., 2023a; XAI, 2024; Liu et al., 2023a; Yu et al., 2024; Marino et al., 2019) Moreover, considering the needs of cloud-side LMMs in the embodied AI, we collected data by operating robots in various indoor, architectural, and street environments. Secondly, we recruited human experts to inspect the images and only retained those marked as in-the-wild. Finally, we used the most accurate Image Quality Assessment (IQA) models Q-Align, TOPIQ, and LIQE (Wu et al., 2023b; Chen et al., 2024a; Zhang et al., 2023) as quality controllers, representing the quality from semantic to pixel levels. If any of the indicators is below a certain threshold, the image is deemed to be distorted and removed. The specific image proportions and processing details are included in the Appendix. Finally, we add question-answer pairs to these images. For samples from other datasets that already included question-answer pairs, we retain them if our human experts could correctly answer. Otherwise, we re-annotate them. We also set new question-answer pairs for all samples we collected ourselves. As a result, we obtained 1,485 question-answer pairs, with 495 samples each for MCQ, VQA, and CAP as ground truth.

### 3.2 DISTORTED DATA COLLECTION

To comprehensively characterize the corruption that images encounter in the real world, we divide the process from image capture to large model reception into seven steps: Environment Interference (EI), Camera Interference (CI), Analog-to-Digital (AD), Source Encoding (SE), Channel Transmis-

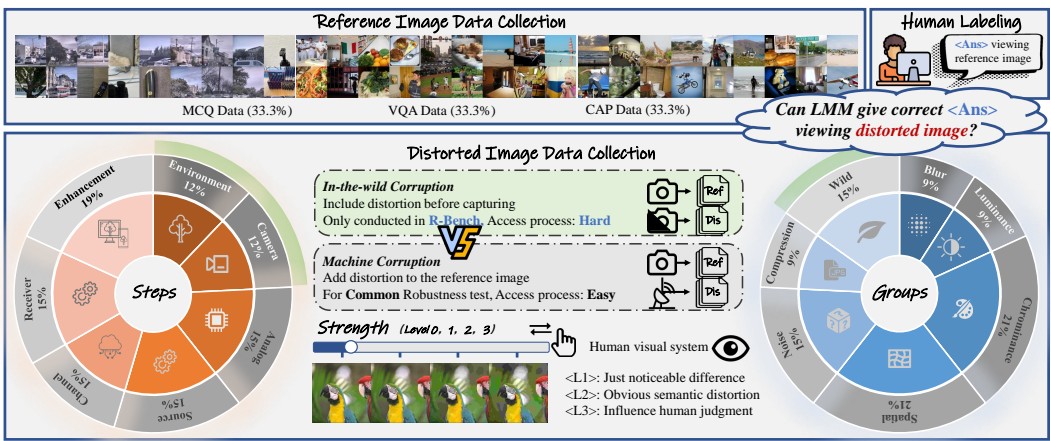

Figure 2: Data collection process of **R-Bench**. We collect and annotate reference image data with 3 different tasks, then construct distorted images with *In-the-wild corruption* by changing the environment before imaging, and *Machine corruption* by adding distortions. All 33 corruption dimensions belong to 7 **steps** (in time order) and 7 **groups** (in low-level), each having three levels of strength.

Table 2: All 33 corruption dimensions of R-Bench, listed by 7 distortion steps. Different icons denote 7 low-level group Blur (○), Luminance (□), Chrominance (△), Spatial (◇), Noise (⋆), Compression (+), and Wild (×).

| Step | Explanation | Dimensions |
|---|---|---|
| Environment Interfere (EI,1) | Interference with the sub-ject to be photographed | 1.Motion blur ○; 2.Bright illumination ×; 3.Dark illumination ×; 4.Blocking obstacle × |
| Camera Interfere (CI,2) | Interference with the photographing equipment | 5.Lens blur ○; 6.Resolution limit ◇; 7.Lens obstacle ×; 8.Lens shaking × |
| Analog-to-Digital (AD.3) | Analog-to-Digital conversion mistake by electronic devices | 9.White Noise ⋆; 10.Color Noise ⋆; 11.Impulse Noise ⋆; 12.Multiplicative noise ⋆; 13.Clock jittering ◇ |
| Source Encoding (SE.4) | Information discarded in the source encoding | 14.Color quantization △; 15.JPEG2000 codec △; 16.JPEG codec △; 17.WEBP codec △; 18.Grayscale quantization ◇ |
| Channel Transimssion (CT.5) | Information lost in channel transmission | 19.Block exchange ◇; 20.Block repeat ◇; 21.Block lost ◇; 22.Block interpolation ◇ |
| Receiver Decoding (RD.6) | Information misinterpreted in the receiver decoding | 23.HSV saturation △; 24.LAB saturation △; 25.Maximum brighten □; 26.Minimum darken □; 27.Mean shift □ |
| Enhancement Postprocess (EP.7) | New corruptions introduced to recover above corruptions | 28.Gaussian filter ○; 29.Color diffusion △; 30.Color shift △; 31.CNN denoise ⋆; 32.Shapness change △; 33.Contrast change △ |

sion (CT), Receiver Decoding (RD), and Enhancement Postprocess (EP). Unlike traditional robustness tests, we are the first to focus on the first two steps, EI and CI, which refer to the in-the-wild corruption encountered during the image capture process. We are not only concerned with the corruption that occurs after the image is captured, due to machine signal processing, transmission, and other issues. Note that in-the-wild corruption is more difficult to obtain than machine corruption. It requires changing environmental conditions and camera parameters in reality after capturing high-quality reference images, rather than applying perturbation strategies directly to the reference images as is done with the last five steps. Specifically, we considered 33 common corruption scenarios in the real world as dimensions for our benchmark. These dimensions can be divided into seven steps as mentioned; or, like past IQA work, they can also be categorized by low-level attributes into seven groups: Blur, Luminance, Chrominance, Spatial, Noise, Compression, and the in-the-wild corruption that R-Bench introduced for the first time. Table 2 shows the steps and groups to which each dimension belongs, as well as the definitions of each step. The definitions of each group are based on KADID-10K. (Lin et al., 2019) As space limits, the specific definitions of the 33 dimensions, as well as visualization examples, are provided in the Appendix.

Note that we also controlled the intensity of corruption, which is beneficial for detecting the robustness of LMMs under different corruption levels, as shown in Figure 2. Based on the perceptual

Figure 3: A comprehensive robustness evaluation example. The combination of absolute and relative robustness avoids misjudgment of chance examples, ensuring the reliability of the R-Bench.

mechanism of the HVS, the corruption is divided into three levels: low: humans can detect the difference between the distorted and reference images, which corresponds to the Just-Noticeable Difference (JND) in signal processing; mid: there is a noticeable semantic difference between the two images, but it does not affect human cognition; High: the corruption is severe enough to mislead humans, such as giving incorrect answers to questions like the number of people or the background objects in the image. Within R-Bench, we strictly controlled the intensity of the corruption when capturing distorted images and manually adding distortion, ensuring an average distribution of low/mid/high samples.

## 3.3 ROBUSTNESS DEFINITION

This section defines robustness mathematically to enable a comprehensive robustness assessment. Robustness can be categorized into absolute and relative aspects. Absolute robustness refers to the performance that LMMs exhibit only on distorted images, while relative robustness is whether the outputs of LMMs are stable between reference/distorted images. Thus, absolute robustness $R_a$ can be simply defined as:

$$R_a = \text{Score}(GT, \text{LMM}(I_{dis})), \tag{1}$$

where function $\text{Score}(\cdot)$ compute the similarity between ground truth answer $GT$ and the $\text{LMM}(\cdot)$ result when viewing the distorted image $I_{dis}$. However, this metric is not comprehensive; for a powerful LMM, its performance on distorted images may significantly decline compared to reference images, but since the baseline of the reference is already high, this does not necessarily lower the appearance of $R_a$. Thus, it can be termed a powerful model but not robust. Therefore, it is necessary to add a relative concept above absolute robustness. Some robustness studies (Zhao et al., 2023; Zhang et al., 2024a) attempt to directly express robustness through the output discrepancy between reference and distorted images. Unfortunately, this evaluation is even more unreasonable and can only be referred to as similarity, rather than robustness. For instance, if an LMM produces incorrect outputs, regardless of viewing the reference or distorted image, if these errors happen to be consistent, then this poor model will receive a perfect similarity score. Thus, we have initially defined relative robustness $R_r$ as follows: **Provided that an LMM can correctly process reference images, if the distorted output is still consistent with the reference**, namely:

$$R_r = \text{Score}(GT, \text{LMM}(I_{ref})) \cdot \text{Score}(\text{LMM}(I_{ref}), \text{LMM}(I_{dis})), \tag{2}$$

where $I_{ref}$ denotes the reference image. R-Bench will calculate the performance of all LMMs individually based on the above two metrics and use the average value for the final robustness ranking.

## 4 EXPERIMENT

### 4.1 BENCHMARK CANDIDATES

R-Bench uses 20 mainstream LMMs for testing. All chosen models have demonstrated excellent performance in past multi-modality understanding benchmarks (Liu et al., 2023c; Wu et al., 2023a;

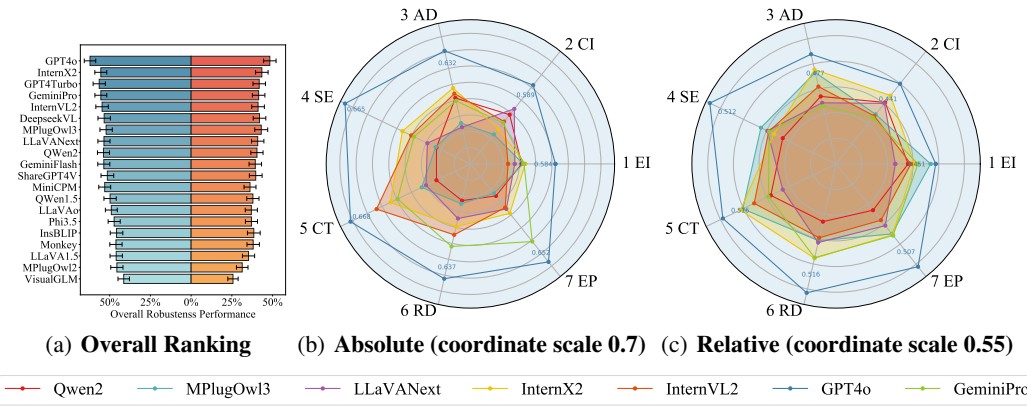

(a) **Overall Ranking**    (b) **Absolute (coordinate scale 0.7)**    (c) **Relative (coordinate scale 0.55)**

Figure 4: Result of R-Bench. (Zoom in to see details) **Absolute** robustness is demonstrated on the left side of (a) and (b); **Relative** robustness is demonstrated on the right side of (a) and (c). Overall the robustness of all LMMs is unsatisfactory, with the proprietary LMMs performing relatively stronger than the open-source. For corruption, Step 1: Environmental Interference and Step 2: Camera Interference has severe negative impacts on all LMMs.

Zhang et al., 2024b) that have relatively strong processing capabilities for reference images, thus ensuring that the robustness results derived by R-Bench are meaningful, including proprietary LMMs: GeminiFlash, GeminiPro (Team, 2024), GPT4o, GPT4Turbo (Achiam et al., 2023); and open-source as 7B-size: DeepseekVL (Lu et al., 2024), InstructBLIP (Dai et al., 2023), InternVL2 (Chen et al., 2024b), InternLM-XComposer2 (Dong et al., 2024), LLaVA1.5 (Liu et al., 2023b), LLaVANext (Li et al., 2024e), LLaVA-OneVision (Li et al., 2024a), MiniCPM (Yao et al., 2024), Monkey (Li et al., 2024f), MPlugOwl2 (Ye et al., 2024b), MPlugOwl3 (Ye et al., 2024a), Phi3.5 (Abdin et al., 2024), QWen1.5-VL (Bai et al., 2023), Qwen2-VL (Wang et al., 2024), ShareGPT4V (Chen et al., 2023), VisualGLM (Du et al., 2022). All LMMs are tested as zero-shot.

Regarding human robustness in the real-world, we conducted a user study in a controlled laboratory environment with five average subjects. They would first view examples of distortions across 33 dimensions and fully confirm that they could understand distorted images. Participants who did not meet the criteria would be excluded. Subsequently, they would watch images from the R-bench in a random order and provide appropriate answers to the questions. Note that for both the LMMs experiment mentioned above and the human subjects here, the MCQ/VQA/CAP tasks were interspersed to prevent any prior knowledge from making the participants familiar with a particular task, thus leading to artificially inflated performance. To avoid the strong resilience to distortions by a few people familiar with photography/communications, the averaged performance of participants except maximum and minimum will serve as the human baseline.

## 4.2 EVALUATION CRITERIA

In the MCQ, VQA, and CAP tasks, we use three LMM-assisted mechanisms as the Score function in Section 3.3. For MCQ, since most LMMs cannot consistently provide instructed output formats, we adopt the following method. If the LMM directly provides options in the form of 'A,B', we directly check the correctness of options; otherwise, we refer to the GPT evaluation proven effective by Liu et al. (2023c) to judge whether the output is semantically consistent with the ground truth. For VQA and CAP tasks, since traditional language metrics like BLEU, CIDEr, and SPICE (Papineni et al., 2002; Vedantam et al., 2015; Anderson et al., 2016) tend to penalize errors rather than reward correct answers, this can lead to advanced LMMs with more complete answers receiving lower scores. (Especially for relative robustness) Therefore, we adopt a similar approach to Q-Bench and A-Bench (Wu et al., 2023a; Zhang et al., 2024b), where a comprehensive score is given to the image based on completeness, precision, and relevance. For VQA, where the target output is less than 10 words, it can be directly compared to the ground truth; for CAP, where the output is around 40 words, we calculate the degree to which each score point in the GT is matched. We repeat the evaluation for each sample five times to avoid chance and collect the weighted average as the final score from 0 to 1. (The MCQ score is binaryized as correct or incorrect.)

Table 3: Results of Absolute (above) and Relative (below) robustness on MCQ/VQA/CAP tasks with 3 corruption strength levels, considering 16 open-source and 4 proprietary LMMs as underlined. The best/second results are marked in Orange/Blue respectively. Long-named models are abbreviated.

| Absolute | MCQ | | | VQA | | | CAP | | | Overall |
|---|---|---|---|---|---|---|---|---|---|---|
| Strength | low | mid | high | low | mid | high | low | mid | high | |
| GPT4o | 0.8176 | 0.7744 | 0.7391 | 0.7184 | 0.7291 | 0.6898 | 0.4235 | 0.4200 | 0.3997 | 0.6348 |
| GPT4Turbo | 0.7059 | 0.6398 | 0.6220 | 0.7055 | 0.7048 | 0.6806 | 0.3698 | 0.3811 | 0.3383 | 0.5722 |
| GeminiPro | 0.7529 | 0.7012 | 0.6708 | 0.6233 | 0.6315 | 0.5796 | 0.4006 | 0.4040 | 0.3734 | 0.5710 |
| InternX2 | 0.7176 | 0.6770 | 0.6220 | 0.6288 | 0.6255 | 0.6180 | 0.4204 | 0.3982 | 0.3659 | 0.5638 |
| InternVL2 | 0.7118 | 0.7019 | 0.6280 | 0.6442 | 0.6436 | 0.6383 | 0.3759 | 0.3669 | 0.3412 | 0.5614 |
| GeminiFlash | 0.7235 | 0.6708 | 0.7073 | 0.5975 | 0.6036 | 0.5575 | 0.3840 | 0.3522 | 0.3487 | 0.5495 |
| LLaVANext | 0.6529 | 0.6087 | 0.5732 | 0.6276 | 0.6382 | 0.6150 | 0.3957 | 0.4006 | 0.3873 | 0.5445 |
| MiniCPM | 0.7081 | 0.6471 | 0.5610 | 0.5626 | 0.6024 | 0.5880 | 0.4025 | 0.4047 | 0.3885 | 0.5405 |
| Qwen2-VL | 0.6765 | 0.6708 | 0.5732 | 0.5914 | 0.6024 | 0.5335 | 0.4142 | 0.4127 | 0.3787 | 0.5393 |
| DeepseekVL | 0.6149 | 0.5824 | 0.5244 | 0.6679 | 0.6227 | 0.6383 | 0.4167 | 0.4043 | 0.3741 | 0.5384 |
| MPlugOwl3 | 0.6706 | 0.6398 | 0.6159 | 0.5920 | 0.5715 | 0.5671 | 0.3728 | 0.3729 | 0.3729 | 0.5307 |
| ShareGPT4V | 0.6273 | 0.5588 | 0.5488 | 0.6227 | 0.6145 | 0.6473 | 0.3716 | 0.3769 | 0.3390 | 0.5229 |
| Qwen1.5-VL | 0.6087 | 0.5765 | 0.5000 | 0.6178 | 0.5642 | 0.5964 | 0.3895 | 0.3717 | 0.3502 | 0.5083 |
| LLaVAo | 0.5353 | 0.5652 | 0.5305 | 0.5387 | 0.5255 | 0.5749 | 0.4259 | 0.3990 | 0.3809 | 0.4972 |
| Phi3.5 | 0.5765 | 0.5652 | 0.5244 | 0.5337 | 0.5679 | 0.5114 | 0.3660 | 0.3564 | 0.3342 | 0.4818 |
| Monkey | 0.5471 | 0.5155 | 0.4451 | 0.5712 | 0.5648 | 0.5413 | 0.3833 | 0.3487 | 0.3573 | 0.4750 |
| LLaVA15 | 0.4706 | 0.4596 | 0.4695 | 0.6049 | 0.5679 | 0.6210 | 0.3457 | 0.3433 | 0.3476 | 0.4701 |
| MPlugOwl2 | 0.5647 | 0.5652 | 0.5000 | 0.5245 | 0.5255 | 0.5311 | 0.3364 | 0.3284 | 0.3043 | 0.4645 |
| InstructBLIP | 0.4529 | 0.5280 | 0.4756 | 0.5534 | 0.5467 | 0.5653 | 0.3284 | 0.3414 | 0.3557 | 0.4606 |
| VisualGLM | 0.4765 | 0.5217 | 0.5061 | 0.3994 | 0.3885 | 0.3623 | 0.3864 | 0.3571 | 0.3830 | 0.4198 |

| Relative | MCQ | | | VQA | | | CAP | | | Overall |
|---|---|---|---|---|---|---|---|---|---|---|
| Strength | low | mid | high | low | mid | high | low | mid | high | |
| GPT4o | 0.7471 | 0.6894 | 0.6159 | 0.5787 | 0.5725 | 0.5622 | 0.2274 | 0.2134 | 0.2083 | 0.4907 |
| InternX2 | 0.6353 | 0.6087 | 0.5488 | 0.5038 | 0.5127 | 0.4639 | 0.2440 | 0.2317 | 0.2070 | 0.4396 |
| MPlugOwl3 | 0.6087 | 0.5882 | 0.5488 | 0.5242 | 0.4877 | 0.4938 | 0.2423 | 0.2106 | 0.2205 | 0.4359 |
| GPT4Turbo | 0.5941 | 0.5590 | 0.4817 | 0.5872 | 0.5575 | 0.5196 | 0.1972 | 0.1910 | 0.1836 | 0.4302 |
| DeepseekVL | 0.5706 | 0.5342 | 0.4756 | 0.5384 | 0.5164 | 0.4934 | 0.2540 | 0.2341 | 0.2089 | 0.4251 |
| GeminiPro | 0.6706 | 0.6211 | 0.5793 | 0.4640 | 0.4799 | 0.4510 | 0.1773 | 0.1874 | 0.1649 | 0.4219 |
| InternVL2 | 0.6294 | 0.6149 | 0.5427 | 0.4849 | 0.4850 | 0.4556 | 0.1940 | 0.1893 | 0.1698 | 0.4185 |
| LLaVANext | 0.5882 | 0.5155 | 0.4817 | 0.5061 | 0.5065 | 0.4531 | 0.2310 | 0.2159 | 0.2161 | 0.4129 |
| Qwen2-VL | 0.6706 | 0.6522 | 0.5244 | 0.4217 | 0.3926 | 0.3627 | 0.2210 | 0.2025 | 0.1957 | 0.4049 |
| GeminiFlash | 0.6235 | 0.5714 | 0.6037 | 0.4397 | 0.4719 | 0.4031 | 0.1681 | 0.1709 | 0.1654 | 0.4021 |
| ShareGPT4V | 0.5528 | 0.4941 | 0.4573 | 0.5067 | 0.4897 | 0.5303 | 0.2109 | 0.2032 | 0.1733 | 0.4019 |
| Monkey | 0.4647 | 0.4534 | 0.3598 | 0.5036 | 0.4882 | 0.4546 | 0.2828 | 0.2451 | 0.2379 | 0.3877 |
| Qwen1.5-VL | 0.5342 | 0.4941 | 0.4085 | 0.4712 | 0.4311 | 0.4637 | 0.2530 | 0.2152 | 0.2046 | 0.3860 |
| InstructBLIP | 0.4529 | 0.4720 | 0.4451 | 0.4937 | 0.4615 | 0.4738 | 0.2343 | 0.2302 | 0.2061 | 0.3855 |
| LLaVAo | 0.5059 | 0.5093 | 0.4817 | 0.4482 | 0.4224 | 0.4214 | 0.2078 | 0.2075 | 0.2012 | 0.3784 |
| Phi3.5 | 0.5176 | 0.4845 | 0.4695 | 0.4564 | 0.4891 | 0.3896 | 0.1915 | 0.1950 | 0.1825 | 0.3751 |
| MiniCPM | 0.5176 | 0.5280 | 0.4512 | 0.3909 | 0.4233 | 0.4223 | 0.1967 | 0.1885 | 0.1955 | 0.3683 |
| LLaVA1.5 | 0.4412 | 0.3727 | 0.3720 | 0.5101 | 0.4558 | 0.5101 | 0.1955 | 0.1883 | 0.1850 | 0.3592 |
| MPlugOwl2 | 0.4529 | 0.4472 | 0.4146 | 0.3480 | 0.3659 | 0.3551 | 0.1676 | 0.1628 | 0.1507 | 0.3184 |
| VisualGLM | 0.3765 | 0.4161 | 0.3841 | 0.2038 | 0.1990 | 0.1722 | 0.1951 | 0.1911 | 0.2071 | 0.2604 |

## 4.3 BENCHMARK RESULT AND DISCUSSION

Figure 4 shows the absolute and relative robustness of 20 LMMs. The 90% confidence intervals of each LMM are marked as error bars, indicating that the scores of the same LMM on different test samples are similar, indirectly proving that the results of the R-Bench test are stable and credi-

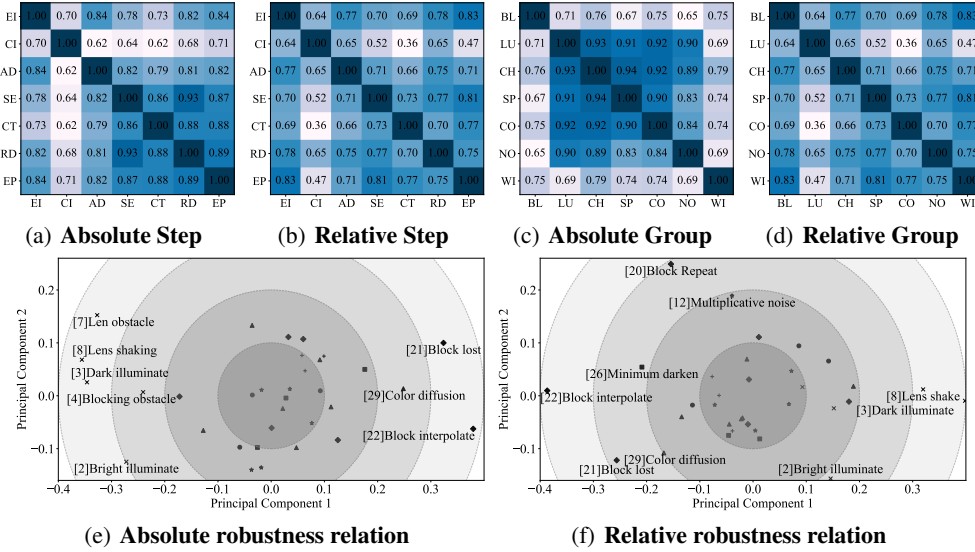

(a) **Absolute Step**   (b) **Relative Step**   (c) **Absolute Group**   (d) **Relative Group**

(e) **Absolute robustness relation**          (f) **Relative robustness relation**

Figure 5: Correlation mat between 7 steps and 7 groups in absolute and relative quality. Principal components of all 33 corruption dimensions are analyzed. Prominent similarity is reflected by higher values from (a) to (d), and closer distance in (e) and (f). The shape of each point obeys Table 2.

ble. Figures (b) and (c) demonstrate that GPT4o is fully superior to other models in each distortion step, with an overwhelming advantage in absolute robustness and a slight lead in relative robustness. The open-source LMMs InternLM-XComposer2 and InternVL2 perform relatively well and can surpass proprietary LMMs (except GPT4o) in some dimensions. Most LMMs score lower in the first two steps, and relatively higher in the last five. This reflects that their training process may have incorporated machine-related distorted images, especially compressed and partially masked ones (which correspond to steps 4 and 5, where LMMs are currently most proficient), thus having some robustness. However, the wild-in-the-distortion data for the first two dimensions needs to be collected manually rather than relying on machine simulation, so LMMs find it difficult to handle this unseen corruption. Table 3 provides a detailed overview of the performance of LMMs under different tasks and corruption levels. Overall, closed-source models dominated the top three positions in terms of absolute robustness, while open-source models exhibited better relative robustness. In terms of tasks, the proficiency order of all LMMs is MCQ>VQA>CAP, indicating that as the output format becomes more complex, corruption becomes more likely to lead to incorrect outputs, which negatively impacts robustness; in terms of corruption level, the higher the level, the worse the LMM's performance. This suggests that LMMs and humans generally share the same preference for distorted images, with only two exceptions. Firstly, some LMMs are most sensitive to the low>mid change, such as GPT4Turbo, yet are unaffected by mid>high corruption; some LMMs exhibit completely opposite sensitivities between the two, like Qwen2-VL. This suggests the 'image quality degradation' perceived by LMMs is not entirely linear. Secondly, we have found in a few cases that LMMs experience an increase in performance after corruption intensifies, such as ShareGPT4V in the VQA task. This indicates that corruption is not always detrimental, and specific corruptions may promote feature extraction in certain models, thereby stimulating the model emergence. (Wei et al., 2022) Both of these interesting findings warrant further investigation.

## 4.4 CORRUPTION ANALYSIS

To explore the relationships between each corruption dimension, we calculated the SRoCC (Spearman Rank-order Correlation Coefficient) for the 7 steps and 7 groups of corruption, based on the performance of LMMs, as shown in Figure 5. The data from (a) to (d) show that the differences in relative robustness between models are greater than those in absolute robustness, with the results of steps such as Camera interfere and categories like Wild differing the most from the others. These dimensions include a large number of in-the-wild corruptions, and this difference also proves the necessity of considering corruption for the first time in the robustness evaluation. (e) and (f) take the performance of the 33 corruptions for a principal component analysis, with most objects distributed

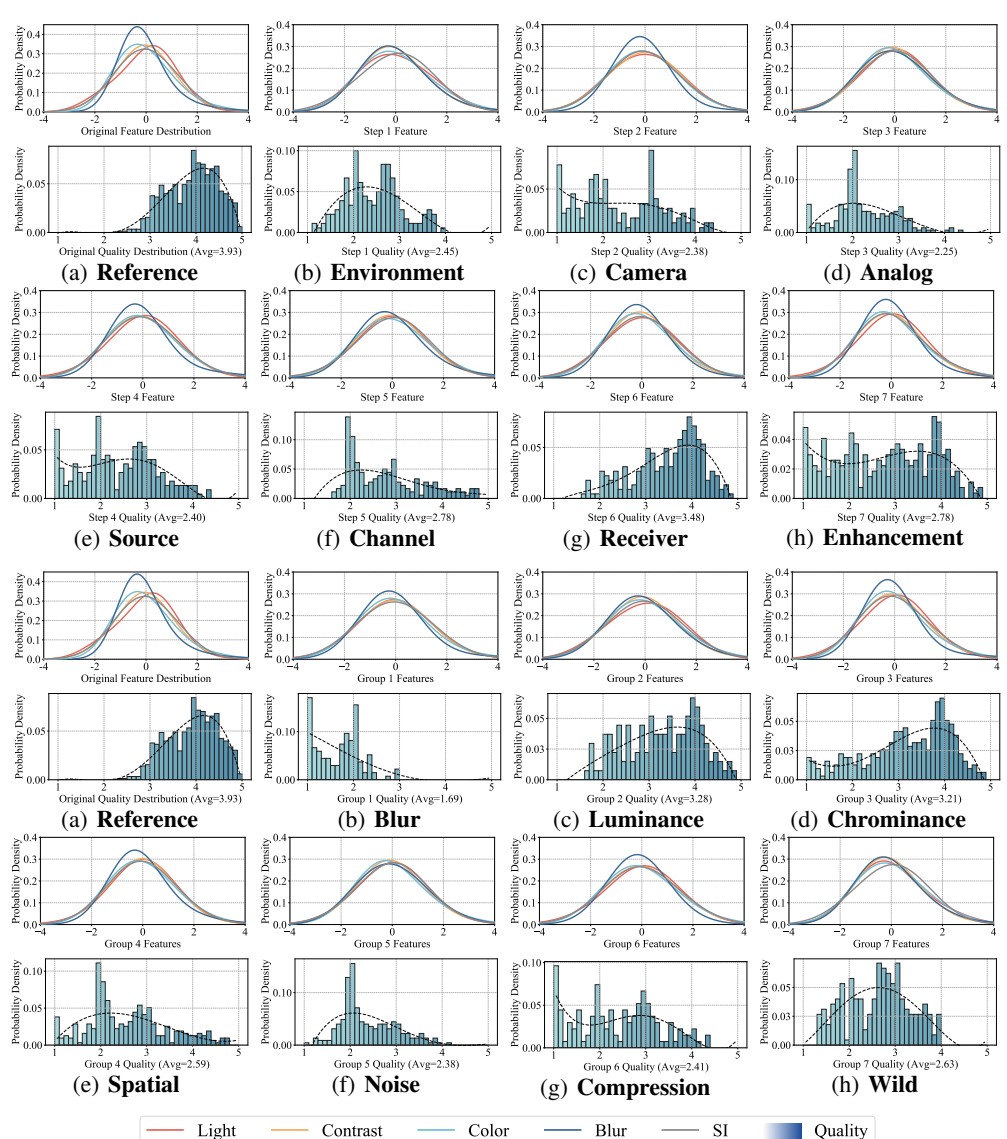

Figure 6: Low-level feature curve and quality distribution of reference/distorted images in different corruption steps/groups. Flatter curves and lower quality are more sensitive to corruption.

relatively close together, and sub-dimensions 2, 3, 8, 21, and 22 differing significantly from the others. Therefore, future robust LMMs need to focus on these aspects.

Figure 6 further analyzes the low-level features of the reference/distorted image in R-Bench.We calculated the distribution of five quality-related attributes, including light, contrast, color, blur, and Spatial Information (SI, representing the content diversity of the image). Detailed explanations of these attributes can be found in work (Hosu et al., 2017). Specifically, the flatter the distribution, the more extreme values are represented, implying such corruption changes the image more. Meanwhile, we calculate the low-level quality distribution for each corruption step and group using Q-Align, (Wu et al., 2023b) and label the mean value on the x-axis. Considering the above two factors together, we find the steps EI/AD/CT, and group Blur/Noise bring the most significant corruption to the image. However, the dimensions that hallucinate LMMs most in Figures 4 and 5, Step: CI, and Group: Wild do not change significantly. Summarizing the above information, we conclude that the results for the perception of image corruption were different at the signal processing level, human subjective level, and the LMM level. Therefore, the perceptual mechanism of LMM will be the key to solving its real-world robustness problem.

Table 4: Absolute robustness comparison between *GPT-4o* and *human* (**left/right**). Evaluated by 3 tasks, 3 strength, 7 steps, and 7 groups. As the R-bench champion, *GPT-4o* still lags behind *human* across the board. **Orange**/**Blue** denote *GPT-4o* performance below 90% or above 98% of *humans*.

| Task | MCQ | VQA | CAP | Strength | low | mid | high |
|---|---|---|---|---|---|---|---|
| | 0.735/0.909 | 0.712/0.678 | 0.414/0.425 | | 0.644/0.671 | 0.615/0.670 | 0.604/0.672 |
| Step | Environment | Camera | Analog | Source | Channel | Receiver | Enhancement |
| | 0.578/0.625 | 0.572/0.602 | 0.614/0.675 | 0.656/0.663 | 0.657/0.713 | 0.620/0.708 | 0.634/0.692 |
| Group | Blur | Luminance | Color | Spatial | Noise | Compress | Wild |
| | 0.604/0.622 | 0.621/0.684 | 0.647/0.693 | 0.651/0.686 | 0.613/0.675 | 0.666/0.684 | 0.533/0.629 |

## 4.5 GPT4O VS HUMAN

Given that the robustness of GPT leads in various downstream tasks, corruption intensity, and across the majority of dimensions compared to all existing LMMs, we compare it with human performance in Table 4. Since answers from humans on reference images are almost GT, there is no statistical difference in absolute/relative robustness. Therefore, we only compare absolute robustness, which is where LMMs are more proficient. Unfortunately, we find that GPT4o still has a significant gap compared to humans, although it achieved a perfect score in the R-Bench evaluation. The only task where GPT4o surpasses human performance is VQA, and the main reason is the openness of the questions, leading to different answers from humans for the same sample, rather than corruption's influence. In addition, only Step: SI reaches 98% of human performance. In other aspects, GPT4o shows a comprehensive disadvantage, especially in MCQ tasks and at high corruption levels; in addition, Steps: RD and Group: Wild are the main sources of the gap.

In summary, based on the above analyses, we believe that current LMMs have some robustness against corruption but are not suitable for the real world. To address the variety and severity of corruption in the real world, further optimization is needed in the following aspects:

- For LMMs: The optimization focus for proprietary LMMs is on relative robustness, ensuring that the output on distorted images matches that of reference images, gradually approaching and potentially surpassing human performance. Open-source LMMs, however, first need to ensure absolute robustness, achieving correct results on reference images, and improving their resilience to corruption only after enhancing their original performance.

- For corruption: In the link from the agent capturing to the LMM perceiving, the first two steps, which are in-the-wild distortions, need to be given special attention. Their negative impact on model robustness far exceeds that of the subsequent five steps related to machines. On the one hand, LMM developers need to use more distorted data for training; on the other hand, current users need to avoid such issues when using LMMs.

- For robustness itself: Assessing LMM robustness is currently the biggest challenge. Experiments show that in some dimensions, there is a significant decline in image quality, yet the performance of LMMs is barely affected; while in other relatively minor distortions, LMMs produce severe hallucinations. In the future, an end-to-end assessment for LMM robustness is needed, explaining the correlation between corruption and LMM perception mechanisms, thereby inspiring LMMs to handle various images in the real world.

## 5 CONCLUSION

We construct R-Bench, a benchmark for evaluating the robustness of LMMs in the real world against corruption, indicated by the performance of LMMs on distorted images in three downstream tasks: MCQ, VQA, and CAP. We fully modeled the pipeline from the agent capturing to the LMM perceiving for the first time, and classified 33 common corruptions in the real world, including 7 time steps and 7 low-level groups with high-quality human annotations. Through our first-ever absolute-relative comprehensive robustness evaluation, we find that proprietary models outperform open-source models but still significantly lag behind humans, which are not yet ready for the real-world. Extensive experimental analysis of corruption also reveals factors that lead to the lack of robustness. We sincerely hope that R-Bench will inspire future LMMs to achieve better robustness, extending their applications from experimental simulations to the real-world.

## ETHIC STATEMENT

The research conducted in the paper conforms, in every respect, with the ICLR Code of Ethics. The data collection, processing, and analysis all comply with the declaration of Helsinki. Official ethical certificates and stamps of approval were obtained before the experiment. Each user provides informed consent for their data to be used in experiments. as shown in Figure 7.

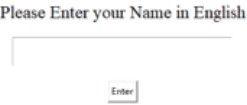

You are being asked to participate in a research study. Before you decide, it is important for you to understand why the research is being done and what it will involve. Please take your time to read the following information carefully and ask questions about anything you do not understand. This form describes a research study that you are invited to take part in.

Purpose of the Study The purpose of this study is to annotate your response towards question-image pairs.

Procedures If you agree to take part in this study, the researcher will collect and use data from your preference. The data may include, but is not limited to, scientific research, subjective analysis, model training.

Risks There are minimal foreseeable risks associated with the use of your data for research purposes. However, as with any data collection and storage, there is a risk of unauthorized access despite all reasonable security measures being taken.

Benefits The potential benefits of this research include 60-80 CNY according to your annotation quality.

Confidentiality Your data will be treated confidentially and will only be accessible to the researcher(s) involved in this study. All identifiable personal information will be kept confidential and will not be shared outside of the research team.

Voluntary Participation and Withdrawal Your participation in this study is voluntary. You have the right to refuse to participate or to withdraw your consent at any time without affecting your current or future relations with the researcher or organization.

I have read the above information, and I have had the opportunity to ask questions and have had those questions answered to my satisfaction. By providing my data and signing below, I consent to participate in this research study and for my data to be used for research purposes.

Please Enter your Name in English

Enter

Figure 7: Data Collection Agreement.

## REPRODUCIBILITY STATEMENT

We have provided implementation details in Sections 4.1 and 4.2. We will also release all the code. The benchmark is a long-term project, which will be updated every month by the R-Bench author team. We look forward to testing the robustness of more advanced LMMs in the future. All users are free to use R-Bench-related resources. If anyone wants to extend the benchmark, including but not limited to Robustness Indicators, new LMMs, and data about different tasks/corruption dimensions can contact us and their contributions will be reviewed.

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

# A APPENDIX

## A.1 LIMITATIONS AND SOCIAL IMPACT

**Limitation 1**: Although R-Bench considers a wide range of LMMs, there are always more advanced models that cannot be taken into account. Especially for the robustness task, experiments have shown that LMMs do not lack the relevant knowledge, but are guided to hallucinate by corruption, thus giving incorrect answers. With the rapid iteration of LMMs, although we regret that we cannot test these upcoming advanced models, we sincerely hope that the current R-Bench can be an assistant for future LMM evolution on robustness.

**Limitation 2**: Although the evaluation of R-Bench is comprehensive and reliable, it still needs assistance from text-modal GPT, which is a common problem of many current benchmarks. (Wu et al., 2023a; Zhang et al., 2024b; Liu et al., 2023c) This limits its usage as a reference list for the model level, rather than an evaluation plug-in for the instance level. For the evolution of LMM, if the image preference of LMM can be obtained in real-time through an open-source pipeline, robustness can be measured through Reference/Distorted image score disparity. This will be very helpful for its understanding and processing of distorted images, thus improving its usability in the real-world.

**Social Impact**: We believe R-Bench can drive innovation by providing a standardized platform for comparing different models and their resilience against corruption. Evaluating how well LMMs handle imperfections can significantly enhance the reliability of real-world downstream tasks, helping them meet certain robustness standards before deployment. This is crucial for applications where accuracy is critical while corruption is easy to occur, such as medical imaging, autonomous vehicles, and embodied AI.

## A.2 CORRUPTION DETAIL

All 33 corruption are listed below, with the (Step, Group) they belong to:

1. Motion blur (EI, Blur): The object itself is moving, causing a blur trail.
2. Bright illumination (EI, Wild): A very bright light source in the environment, interfering with the imaging.
3. Dark illumination (EI, Wild): No light source in the environment, making it difficult to image.
4. Blocking obstacle (EI, Wild): An obstacle blocks part of the object being photographed.
5. Lens blur (CI, Blur): Lens fog causes refraction.
6. Resolution limit (CI, Spatial): The lens resolution is insufficient, requiring upsampling.
7. Lens obstacle (CI, Wild): An obstacle blocks part of the lens.
8. Lens shaking (CI, Wild): The camera shakes randomly in the direction of movement, making it difficult to capture clear images.
9. White Noise (AD, Noise): Overall aging or prolonged operation of the circuit causes noise.
10. Color Noise (AD, Noise): Damage to certain channels in the YCbCr of the circuit components.
11. Impulse noise (AD, Noise): External factors cause sudden interference to the circuit components.
12. Multiplicative noise (AD, Noise): The power supply voltage does not match the required voltage of the circuit components.
13. Clock jittering (AD, Spatial): The frequency of the clock module is inaccurate, causing frequency oscillation.
14. Color quantization (SE, Chrominance): Similar colors are merged through minimum variance quantization.
15. JPEG2000 codec (SE, Compression): A standard image compression method.
16. JPEG codec (SE, Compression): The most widely used image compression method.
17. WEBP codec (SE, Compression): The image compression method with the best comprehensive performance.
18. Grayscale quantization (SE, Spatial): 256 colors are mapped to fewer dimensions through uniform quantization.
19. Block exchange (CT, Spatial): The order of two Macro-blocks in the channel is wrong.
20. Block repeat (CT, Spatial): A Macro-block is convoluted twice and covers the original position.
21. Block lost (CT, Spatial): A Macro-block is lost, and some communication protocol turns it into random pixels.

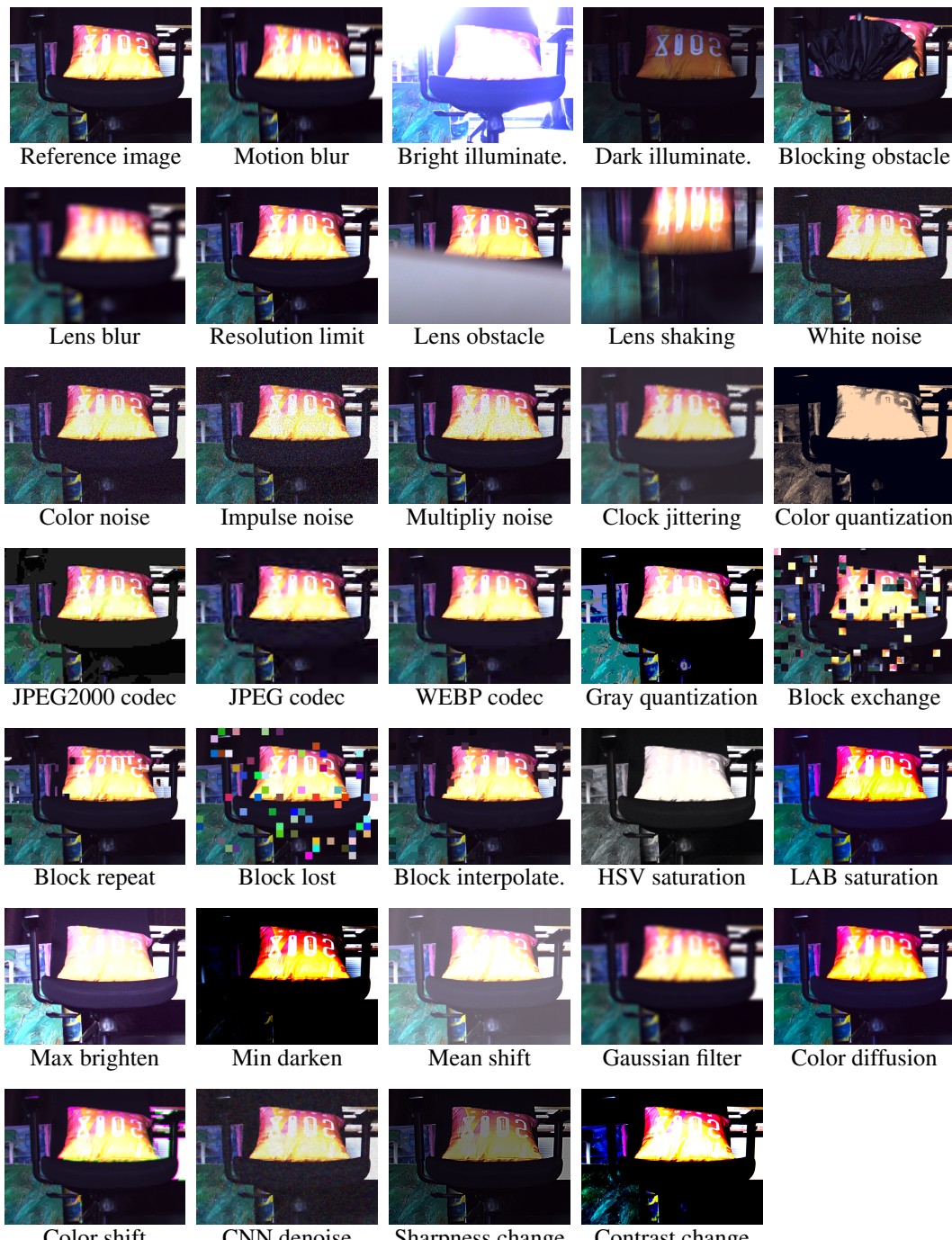

Figure 8: Visualization example of the reference image and its all 33 corruption example. Corruption names will be abbreviated if too long.

22. Block interpolation (CT, Spatial): A Macro-block is lost, and some communication protocol uses surrounding pixels to interpolate it.

23. HSV saturation (RD, Chrominance): The saturation channel of the HSV image is incompatible with the decoder.

24. Lab saturation (RD, Chrominance): The color channel of the Lab image is incompatible with the decoder.

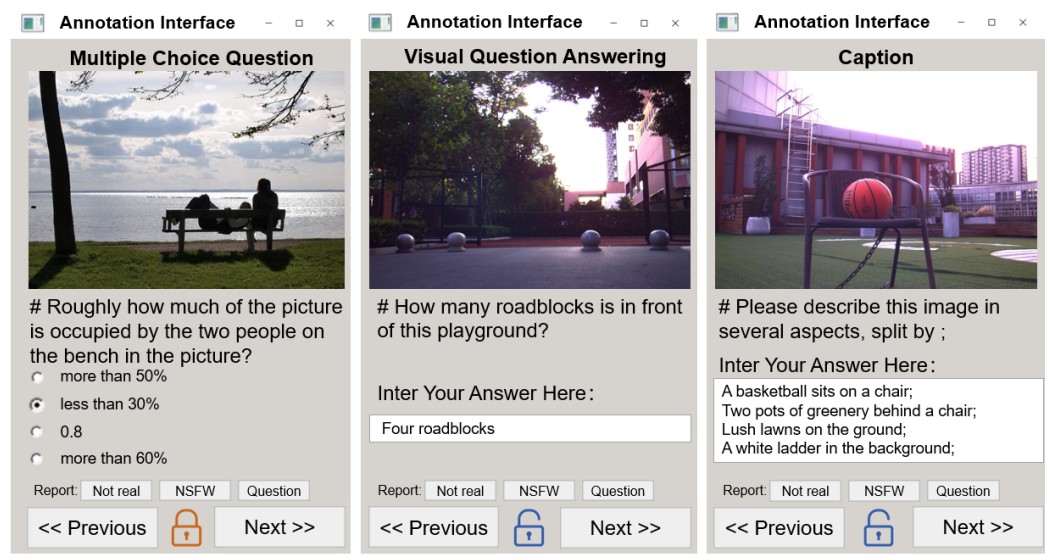

Figure 9: Human user interface for labeling. Three tasks will come in random order.

25. Maximum brighten (RD, Luminance): Some non-linear decoders keep the maximum value unchanged and increase other pixel values.

26. Minimum darken (RD, Luminance): Some non-linear decoders keep the minimum value unchanged and decrease other pixel values.

27. Mean shift (RD, Luminance): Some decoders automatically adjust contrast, but sudden changes in lighting are not yet adapted.

28. Gaussian filter (EP, Blur): New noise introduced by removing grainy salt-and-pepper noise.

29. Color diffusion (EP, Chrominance): New noise introduced by removing certain salt-and-pepper noise in one channel.

30. Color shift (EP, Chrominance): Unreasonableness introduced through the recovery of another missing channel through a certain channel.

31. CNN denoise (EP, Niose): AI-artifacts introduced through neural networks in image-reconstruction or super-resolution tasks.

32. Sharpness change (EP, Chrominance): Over-sharpening caused by excessive configuration of the sharpness by some users.

33. Contrast change (EP, Chrominance): Details lost due to excessive configuration of the contrast by some users.

The example of all corruption is shown in Figure 8, for clear visualization, the corruption strength is set as 'high'. Overall, the content of each Step and Group is relatively homogeneous, with none of them containing too many or too few sub-dimensions, justifying the categorization.

### A.3  USER INTERFACE

The user annotation interface is shown by Figure 9, where subjects will complete interspersed MCQ/VQA/CAP tasks in a randomized order, with some samples being annotated and others not. Subjects can make the following decisions:

- If they agree with the labeling, then click Next;

- If they do not agree with the annotation, they click Unlock to get permission to re-edit the content;

- If the question itself does not make sense, click Question in Report, and the image will be sent to the R-Bench expert team to redesign the question;

- If seeing an unnatural image, or NSFW content is found, click the corresponding button in Report, and this sample is excluded from R-Bench.

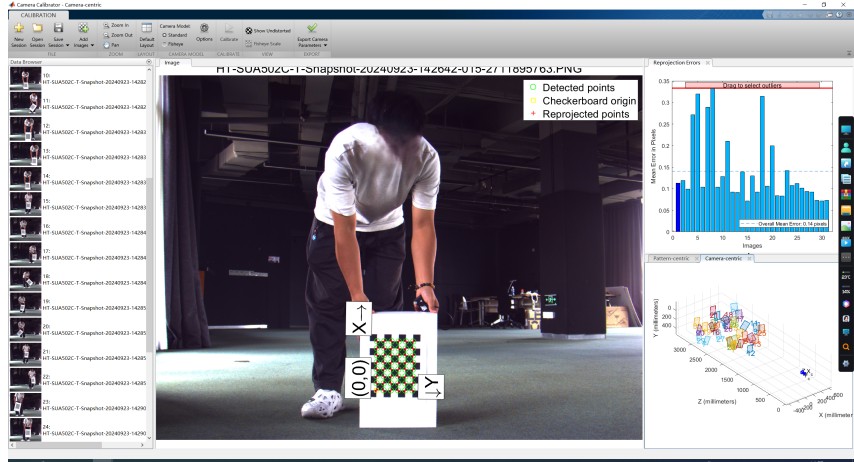

Figure 10: Camera parameter labeling with the mosaic board. The expert face is masked for privacy.

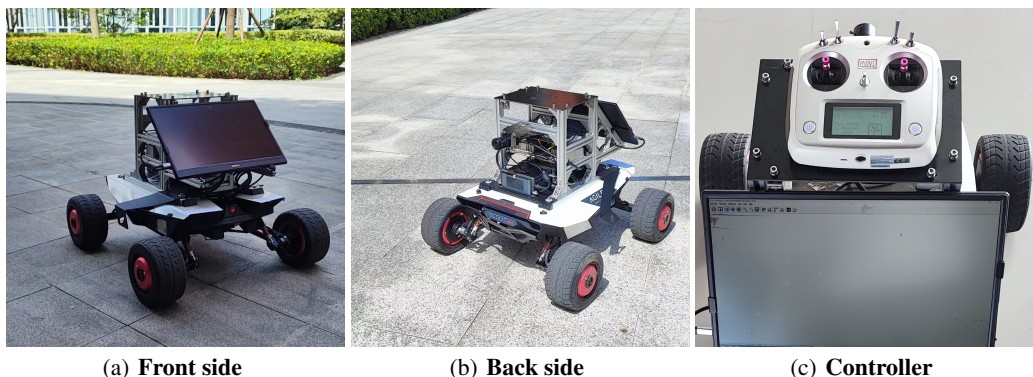

(a) **Front side**  (b) **Back side**  (c) **Controller**

Figure 11: The final assembled robot.

The word limits for VQA and CAP tasks are 10 and 40, and we do not encourage users to write too long content. All labeled content will be reviewed by the R-Bench team to eliminate low-quality data and reserve high-quality data from 5 subjects, thus ensuring the reliability of the benchmark.

### A.4 IMAGE COLLECTION PROCESS

The R-Bench author team has a rich cross-disciplinary background, including computational photography, bit-attitude estimation, robot manipulation, source/channel coding, and many other techniques. Together, they ensured a smooth implementation of R-Bench. The data outside of the in-the-wild in the 33 corruption dimensions was handled by the channel group. They are responsible for manually collecting the data and simulating the distortion using Matlab. The more difficult in-the-wild data is taken care of by the robotics group, i.e., the reference/distortion images are collected in the same scene. Therefore, the robotics group needs to make sure that both the first image taken is of high quality, and the second image needs to be identical to the first one in terms of scenery except for the target distortion. In order to shorten this interval, the camera needs to be fully set up in advance.

We first assembled the camera with the robot. The camera is HUATENGVISION HT-SUA502C and the robot is Agilex Scout Mini. At this point, due to some deviation in the camera position, a correction is required to ensure the quality of the reference image.

Then, the camera calibration is performed, we use the classic mosaic calibration plate to calculate the degree of offset of the bit position, from which the initial parameters of the camera are adjusted, as shown in Fig. 10. The result is listed as:

$$Focal\_Length(fx, fy) = [3460.7560, 3391.3461]$$

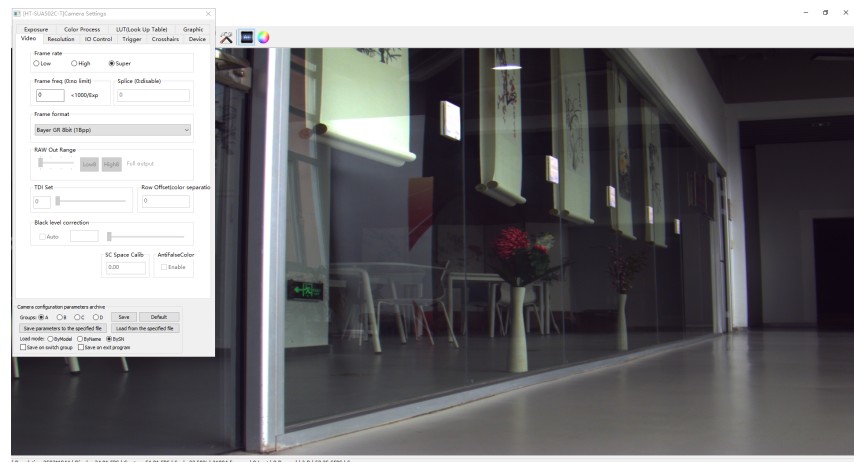

(a) **Robot see before distortion**

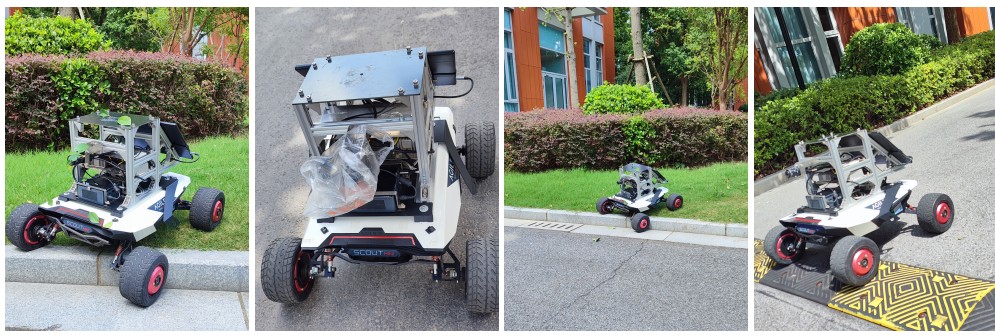

(b) **Actual lens obstacle** (c) **Actual lens obstacle** (d) **Actual lens shaking** (e) **Actual lens shaking**

Figure 12: Robot before encountering corruptions.

$$Principal\_Point(cx, cy) = [1270.9, 990.0620]$$
$$Radial\_Distortion = [-0.1235, 0.2016]$$

The final assembly is shown in Fig. 11. Next, we manipulate the robot to collect samples in a variety of environments, including indoors, in the field, in buildings, on streets, and in garages. We will first take a high-quality image and then immediately manipulate the robot to introduce corruption and take a second distorted image. Figure 12 illustrates this process completely taking dimension 7 lens obstacle and 8 lens shaking as examples. Finally, R-Bench's team of experts will review the two images to see if the difference is large enough and exclude it if it is not significant.

In summary, it can be seen that reference/distorted image pairs are much more difficult to acquire in the in-the-wild condition than machine corruption. However, considering its significance for LMM robustness, we have taken it into account for the first time. We sincerely hope that more future work will consider this dimension to improve the LMM's resistance to distortion through carefully collected image pairs and further generalize its application in embodied intelligence.

## A.5 FINANCIAL DISCLAIMER

Due to financial constraints, we can only try to be as comprehensive as possible in our academic content, such as the corruption dimension, and are not able to consider Hall cameras and robots on the market. This may lead to unavoidable limitations in the content. At the same time, since we do not have access to some of the LMM's temperature parameters, the overly arbitrary play of the output may lead to its unsatisfactory R-Bench ranking. The R-Bench here is only an academic inquiry, not directed at any LMM developer, and does not involve any economic-related rankings. Anyone is welcome to retest their model against R-Bench if needed. We will be happy to update it on the list.

