# OpenReview forum: "R-Bench: Are your Large Multimodel Model Robust to Real-world Corruption?"
_ICLR.cc/2025/Conference — ICLR 2025 Conference Withdrawn Submission_

### Official Review · Reviewer_WsRT · 2024-10-28

**Soundness:** 2
**Presentation:** 3
**Contribution:** 3
**Rating:** 6
**Confidence:** 3

**Summary:**

This paper introduces R-Bench, a benchmark with focus on the robustness of LMMs (Large Multimodal Models) under real-world corruptions. The authors propose a novel Relative Robustness ($R_{r}$) to address the limitations of the existing metrics, particularly the Absolution Robustness ($R_{a}$). R-Bench evaluates LMM performance from comprehensive perspectives, including 33 corruption dimensions spanning 7 steps and 7 groups on low-level visual attributes, as well as a reference/distorted image dataset with 2970 QA pairs. Extensive experiments are conducted on 20 mainstream LMMs across three various tasks, including Multiple Choice Questions (MCQ), Visual Question Answering (VQA), and Captioning (CAP). This benchmark shows a promising direction in advancing LMM evaluation in real-world situations.

**Strengths:**

**S1**: Overall, the authors have provided extensive experiments with detailed discussions.

**S2**: The authors propose a new metric, Relative Robustness ($R_{r}$), with the aim of addressing the limitations in existing benchmarks.

**S3**: R-Bench covers more dimensions than other benchmarks such as MMC-Bench 4+ in more tasks (MCQ, VQA and CAP) and more assessments for Absolute and Relative Robustness.

**S4**: Figures are presented in high quality.

**S5**: Practical guidelines and takeaways are provided for practitioners regarding both training and usage, as well as LMM evaluation.

**S6**: Comparison with human performance is presented in Section 4.5 to show the gap in real-world robustness.

**S7**: The proposed benchmark R-Bench has the potential to evaluate robustness for LMMs, which can facilitate further LMMs robustness improvements.

**Weaknesses:**

**W1**: Since the Relative Robustness ($R_{r}$) is the main proposed metric aimed at addressing the limitations of existing evaluations, the major issue is that **it is unclear whether the advantages of the proposed metric ($R_{r}$) in R-Bench over existing metrics, particularly Absolute Robustness ($R_{a}$), are demonstrated through experimental results.**

While the authors have provided the conceptual advantages of ($R_{r}$) in Figure 3 with an illustrative example, as well as comprehensive results (Fig. 4-6, Table 3) for 20 LMMs with detailed discussions (which is appreciated), it is crucial to justify how and to what extent $R_{r}$, or the combination of $R_{r}$ and $R_{a}$ is more effective than $R_{a}$ alone in capturing LMM robustness against real-world corruption by experiments, not just by illustrative conceptual examples. Further experiments are encouraged as follows:

**W1.1**: demonstrate when and why $R_{a}$ cannot capture certain aspects of robustness mentioned in Figure 3 (four combinations of Reference/Distorted), analyze the performance across the three LMM tasks, and provide case studies (such as distributions or concrete examples for each robustness), or to conduct a correlation analysis between $R_{r}$ and robustness or real-world performance. Figure 1 contains nice illustrative examples but the authors are encouraged to provide quantitative results from the dataset.

**W1.2**: demonstrate how $R_{r}$ can help resolve this limitation by revealing the missing aspect of robustness through detailed comparisons. It would be beneficial to quantify these differences as well as some qualitative examples that could be in the Appendix. Without these key comparisons, the findings may be seen as another comprehensive result, making it difficult to substantiate the proposed $R_{r}$’s advantage as stated in the third contribution “...we have proposed the concepts of absolute/relative robustness…”

**W1.3**: Subsequently, in Table 3, the rankings differ from $R_{r}$ (GPT4o, InternX2, MPlugOwl3, …) and $R_{a}$ (GPT4o, GPT4Turbo, GeminiPro, …). The authors have analyzed the differences with detailed discussions (Line 408-421). On Line 262, it says “R-Bench will calculate the performance of all LMMs individually based on the above two metrics (Relative and Absolute Robustness) and use the average value for the final robustness ranking.” The Overall Ranking is summarized in Figure 4 (a) that incorporates $R_{r}$, the authors are encouraged to justify why this ranking is better than the one provided by $R_{a}$ since $R_{r}$ is the newly proposed robustness metric.

**W2**: Missing $R_{r}$ in Section 4.5 GPT4o VS Human. My interpretation of this omission is twofold: (1) LMMs are more proficient in $R_{a}$, if the gap (LMMs vs Human) is already shown to be large in $R_{a}$, it will be even larger in $R_{r}$, so there is no need to conduct further experiments; (2) human answers are almost equivalent to Ground Truth, suggesting no statistical difference. However, since one main contribution is the proposed $R_{r}$, its absence renders the results for R-Bench seem incomplete. It would make the work more complete to include $R_{r}$ results, or to provide more convincing arguments for its omission.

**In summary of weaknesses**, although the comprehensive and experimental provide valuable insights (e.g. the incorporation of machine-related distorted images in the training process), **the role of the proposed $R_{r}$ in supporting $R_{a}$ remains unclear by experimental evidence** in Fig. 4-6 or Table 3.

These comprehensive experiments and analyses result in an overall rating of 6 marginally above the acceptance threshold. However, the identified weaknesses relate to the central claim of the proposed Relative Robustness, the lack of experimental results demonstrating how $R_{r}$ is better than the existing metrics, especially $R_{a}$, prevents a higher score. It would be beneficial for the authors to provide further clarity regarding any key messages that may be missing from this review.


**Presentation**

**P1**: Terminology Consistency: Distortion vs Corruption. This might be a little bit strict on terminology: the term “corruption” is used as a general term to describe the visual attribute degradation of an image in the whole paper, while “distorted image” is also commonly used. Do “corruption” and “distortion” mean exactly the same thing or is “distortion” a specific way of “corruption”? These two terms are used in the whole paper, it seems to me they are used interchangeably to refer to the same thing. Although both have similar meanings, they focus on different aspects especially when it comes to vision - “Distortion” is more about altering the shape or appearance while “Corruption” relates to data integrity or degradation.

Take the visualizations in Figure 8 as concrete examples, “Block Exchange“ or “Block Lost“ are types of corruption, yet most of the text uses “distorted image”. It is unclear if this term refers to the exact same types of corruption listed in Figure 8. It would make it clear if the authors can provide both definitions if they have distinct meanings, or further clarification if these terms are being used interchangeably.

**P2**: Line 251 “...but since the baseline of the reference is already high, this does not necessarily lower the appearance of $R_{a}$.” $R_{a}$ is the absolute robustness defined as $Score(GT,LMM(I_{dis}))$, however, the authors’ intended meaning of the phrase “lower the appearance of the absolute robustness” is unclear. It seems that the authors intended to convey that the value of $R_{a}$ may appear high, yet fails to capture true robustness (false high). A rephrasing for greater clarity could be helpful.

**P3**: Line 400 “Figures (b) and (c) demonstrate that GPT4o” seems to refer to Figure 4, following the context of text, but it is placed next to Figure 5 that also has sub-figures (b) and (c). This placement is initially confusing. It would be clearer to explicitly mention Figure 4 when referring (b) and (c) or put the text and figure closer.

**P4**: Line 404 “This reflects that their training process may have incorporated machine-related distorted images, especially compressed and partially masked ones (which correspond to steps 4 and 5, where LMMs are currently most proficient), thus having some robustness”. This is a nice summary and it would be beneficial if the specific references to these LMMs training process are provided for practitioners.

**P5**: It is unclear to me what the cyan and orange colors represent in Figure 4 (a) until later in the text on Line 376. It might be beneficial to put them closer (e.g. in the caption) for easier reference.

**P6**: In Figure 6, the “Quality” label in the legend is blue, while the corresponding bars in the chart appear cyan, making it unclear whether the brightness of the bar chart represents some values. No explanation is provided in the caption or text. Also, the way to quantify “Quality” is unclear, no formal definition is provided. This leaves room for multiple interpretations, such as subjective human scoring or metrics like PSNR, SSIM, etc, but these are absent from the paper. It would be great if the authors can provide a clear definition of how “Quality” is measured, as well as including the specific metric or method used to quantify “Quality” in the figure caption or main text.

**Minor Issues (Typos or Grammatical Errors)**

**M1**: Figure 3 caption - “3. LMM are able to … 4. LMM are able to” are -> is or LMM are -> LMMs are?

**M2**: Figure 3 caption - “But corruption make it correct as coincidence.” may be corrected as “But the corruption makes it appear correct by coincidence.” (False Positive)?

**M3**: Line 212 The reference mark is placed after the period - “...on KADID-10K. (Lin et al., 2019) As space limits…”.

**M4**: Line 475 “Figure 6 further analyzes the low-level features of the reference/distorted image in R-Bench.We cal…” There is no space between two sentences.

**Questions:**

**Q1**: How is the score computed (e.g. score(Clear,Sunny)) in Figure 3? Although it is described in Equation (1) and in Line 248 - “function Score(·) compute the similarity between ground truth answer GT and the LMM(·)” - it remains unclear to me. It would be great to provide the specific mathematical formulation of score(). Is it based on cosine similarity or inverse distance (such as Euclidean Distance)? Is it computed in a latent space, and if so, which pre-trained model is used? Providing these details seems necessary as the score can vary dramatically using various similarity functions.

**Q2**: Figure 1 caption - “Experiments demonstrate that the LMMs solve the original image but hallucinate against corruption”. What does “solve the original image” mean from a technical perspective? It seems to imply that LMMs can solve a specific task (e.g. MCQ, VQA or CAP) by providing decent answers, given the original image reference. It is difficult to translate “solve the original image” to a specific technical meaningful action.

**Q3**: Line 503 “...it achieved a perfect score in the R-Bench evaluation.” It is unclear which perfect score the authors are referring to, as no “perfect score” (e.g. 1.0) has been presented in the previous results. It seems that the authors intend to convey “best score” or “best performing model”. Further clarification would be beneficial.

**Details Of Ethics Concerns:**

The Ethics Statement states that this research complies with the ICLR Code of Ethics. Users’ personal information during the data collection of the question-pair dataset is confidential.

---

### Official Review · Reviewer_fKx9 · 2024-11-03

**Soundness:** 2
**Presentation:** 3
**Contribution:** 2
**Rating:** 3
**Confidence:** 5

**Summary:**

The paper proposes a benchmark for multi-modal LLMs (here called LMMs): R-Bench includes 33 types of corruption, organised into seven sequences and groups, along with a dataset of reference and corrupted images with 2,970 labelled question-answer pairs for evaluation. LMMs perform well on clean images, but their robustness against distorted images falls short compared to humans, highlighting the need for improvement in real-world applications.

**Strengths:**

The paper addresses an important problem. The low robustness of perception models concerning common, real-world image corruptions/degradations/nuisance shifts is an issue that needs to be studied and improved.

The proposed benchmark is a combination of existing samples from other benchmarks and newly collected images and annotations. The new samples ensure that large VLMs have not yet been trained on this data, resulting in a fair test performance.

**Weaknesses:**

## Positioning
The research community has been investigating model robustness for many years before the advent of VLMs/LMMs. The discussion of the related work and the positioning of the paper are missing a large portion of this work. For example:

Barbu et al., 2019; ObjectNet: A large-scale bias-controlled dataset for pushing the limits of object recognition models

Recht et al., 2019; Do ImageNet classifiers generalize to imagenet?

Wang et al., 2019; Learning robust global representations by penalizing local predictive power

Hendrycks et al., 2021a; The many faces of robustness: A critical analysis of out-of-distribution generalization

Hendrycks et al., 2021b; Natural adversarial examples

Zhao et al., 2022; A benchmark for robustness to out-of-distribution shifts of individual nuisances in natural images

Geirhos et al., 2022; ImageNet-trained CNNs are biased towards texture; increasing shape bias
improves accuracy and robustness

Idrissi et al., 2022; Imagenet-x: Understanding model mistakes with factor of variation annotations

## Findings
A discussion of the contribution of this paper in light of this related work is important to position the paper and to understand which findings are new.
Of course, many of the earlier works target different tasks, such as image classification or representation learning, but the findings are the same: better models are more robust, larger models are more robust, more data helps, etc.

## Robustness Definition
The paper uses a simple score for robustness: multiply the performance score of the clean image with the one from the corrupted image. This has some known disadvantages: it mainly entangles model performance with robustness. Many attempts have been made to formalise and benchmark robustness. An overview can be found in Drenkow et al. (2021) “A Systematic Review of Robustness in Deep Learning for Computer Vision: Mind the gap?”
It would be important to understand why the proposed metric was chosen in favour of more sophisticated ones.

**Questions:**

In addition to the weaknesses above, several questions remain after reading the submission.
* Will the benchmark be released? The paper only mentions releasing the code.
* With new benchmarks, it is very helpful to show many examples of images in the supplementary material so that the reader can get an impression of the breadth of the dataset. The current version of the paper only contains a very small amount of examples and only one sample with many corruptions.

---

### Official Review · Reviewer_LVbr · 2024-11-03

**Soundness:** 3
**Presentation:** 2
**Contribution:** 3
**Rating:** 5
**Confidence:** 4

**Summary:**

The paper introduces R-Bench, a benchmark aimed at evaluating and enhancing the robustness of large multimodal models (LMMs) against real-world corruption. R-Bench simulates the corruption process, from user image capture to LMM processing, and includes 33 corruption dimensions, providing a dataset with 2,970 human-labeled question-answer pairs. The benchmark results indicate that while LMMs perform well on original images, their performance declines significantly with distorted images, highlighting a robustness gap compared to human visual perception.

**Strengths:**

1. The paper presents a comprehensive pipeline to assess LMM robustness against real-world corruption, covering both data collection and evaluation metrics.

2. It introduces 33 corruption dimensions, grouped into 7 categories, including in-the-wild corruptions like environmental and camera interference.

3. The study analyzes model robustness across different corruption types and identifies areas needing improvement, providing useful insights.

**Weaknesses:**

1. The paper lacks a discussion on improving robustness against machine-related distortions through fine-tuning with simulated data. It should provide a baseline to determine whether improvements are easily achievable.

2. Details on dataset collection, especially for in-the-wild corruptions, are insufficient.

3. The definition of corruption levels, particularly the "high" level, is questionable. If corruption is so severe that it misleads humans, is it meaningful to explore robustness under such conditions?

4. The paper does not address how image enhancement methods like SUPIR[1] and Real-ESRGAN[2] might affect LMM performance.

5. The analysis of the gap among the perception of image corruption at the signal processing level, human
subjective level and the LMM level is not enough.

[1]. Yu, Fanghua, et al. "Scaling up to excellence: Practicing model scaling for photo-realistic image restoration in the wild." Proceedings of the IEEE/CVF Conference on Computer Vision and Pattern Recognition. 2024.

[2]. Wang, Xintao, et al. "Real-esrgan: Training real-world blind super-resolution with pure synthetic data." Proceedings of the IEEE/CVF international conference on computer vision. 2021.

**Questions:**

please refer to the weaknesses.

---

### Official Review · Reviewer_V8ga · 2024-11-04

**Soundness:** 2
**Presentation:** 4
**Contribution:** 2
**Rating:** 5
**Confidence:** 3

**Summary:**

This paper proposes a dataset branch mark for the evaluation of the robustness of current major vision LLMs. Specifically, they collect ~3K images and add manual distortion, e.g., blur, noise, and compression. Then they evaluated most of the SOTA LLMs on the introduced dataset (but without a human baseline) for three tasks, i.e. MCQ, VQA, CAP. They also conduct statistics showing the limited robustness of current vision LLMs.

**Strengths:**

1. The experiment is comprehensive and converges almost all SOTA vision LLMs.

2. The presentation is clear and nice. The demonstration of statistics is straightforward and easy to read.

3. The proposed branch mark is considered useful for the comparison of vision LLM quantitatively.

**Weaknesses:**

1. There might be a reproduction issue, that the failure cases might not always occur for vision LLM, the author needs a error bar for each experiment to demonstrate how large the variance is. The examples are included in the next Question sections.

2. The question might be ambiguous. For example, in the very left image of the teaser, I would say there are only two signs in the picture since the top one is a text instruction instead of a sign (when put the image to ChatGPT 4o, it also selected B, which is reasonable from my point of view).

3. 3K image dataset might be too small to cover the variance of real-world object/scene distributions.

4. It is necessary to include single human performance as a baseline in Table 3.

**Questions:**

1. I try to screenshot the corruption examples in the teaser figure and feed them to ChatGPT 4o. But cannot reproduce the issue. Instead, all three answers seem reasonable to me. It is important to generate the corrupted image multiple times and feed it to LLMs to analyze randomness and avoid adversarial examples generated by chance.

a) `B. 2 signs. There are two signs mounted on the same pole in this image.` *why?* `The image shows two distinct sign panels mounted on a single pole, one above the other: Top sign panel: This panel contains information about specific parking or permit requirements.
Bottom sign panel: This panel has a "No Parking" symbol with additional details about restrictions or timings.
While they are on the same pole, each panel conveys separate information, making it count as two signs.`

b) `The green logo on the car in the image is the Monster Energy logo. It's recognizable by its three jagged green "claw" marks, which resemble the letter "M." This logo represents the energy drink brand, known for its association with motorsports and extreme sports events.`

c) `The image shows a blurred, abstract indoor scene with soft lighting, rectangular light reflections, a vertical red structure, and a green exit sign, creating a dreamlike, motion-blurred effect.`

2. What is the instruction used when prompting LLMs? I would expect a template output is passed as the instruction for the LLMs. Otherwise, if you just ask the question (e.g. Please describe this image in detail.), the output of LLM will be to long and complex.

**Details Of Ethics Concerns:**

They include a discussion of ethics concerns.

---

### Official Review · Reviewer_du5u · 2024-11-08

**Soundness:** 3
**Presentation:** 3
**Contribution:** 2
**Rating:** 5
**Confidence:** 4

**Summary:**

The paper introduces a dataset called R-Bench for benchmarking the robustness of large multimodal models (LMMs) against real-world corruptions. Unlike existing benchmarks, which typically focus on adversarial attacks or machine corruptions, the paper highlights that R-Bench is the first to address in-the-wild corruptions, such as environmental illumination interference and obstacle occlusion. Additionally, the paper presents a relative robustness metric that jointly considers the correctness and self-consistency of LMMs. Through extensive experiments on 20 LMMs across three tasks using R-Bench, the paper provides some insights into further optimizing LMMs to withstand real-world corruptions.

**Strengths:**

1.	The paper is well-structured and easy to follow. The included figures and tables, such as Figure 1 and 2, are especially clear and helpful for understanding the paper’s core concepts.

2.	The paper addresses an important general problem: how can we apply LLMs more effectively to real-world scenarios?

3.	The defined corruption steps, groups, and dimensions are comprehensive, covering the entire pipeline from image capture to LMM analysis.

4.	To study in-the-wild corruptions, the proposed R-Bench includes newly collected data --- both clean and corrupted --- obtained by operating robots in various environments, a process known to be relatively challenging.

5.	The LLMs examined in the paper, such as GPT4o and Qwen2-VL, are representative.

**Weaknesses:**

Major:

1.	The reasons provided by the paper (Section 3 and Figure 3) for introducing a new metric (relative robustness, as shown in Equation 2) are not convincing:

To assess the robustness of LLMs when ground truth is available, why not simply present the difference between Score(GT, LMM(Iref)) and Score(GT, LMM(Idis)), which would be more natural and straightforward? As the paper mentions, incorporating the self-consistency metric Score(LMM(Iref), LMM(Idis)) may be unreasonable, as a poorly performing model that produces consistently incorrect outputs could still receive a high score.

The paper does not provide real examples or discussions demonstrating what the proposed relative robustness metric captures that a typical absolute metric does not. From Table 3 and Figure 5, although there are value differences, the general trends captured by the two metrics do not seem significantly different. Additionally, the two metrics might inherently capture similar aspects by definition. Specifically, high relative robustness = high similarity between GT and LMM(Iref) + high similarity between LMM(Iref) and LMM(Idis) ≈ high similarity between GT and LMM(Idis) = high absolute robustness.

2.	Although the paper emphasizes that a major contribution of R-Bench is the inclusion of in-the-wild corrupted data collected by the authors, it appears that this data constitutes only a relatively small portion (15%) of the entire dataset, as shown in Figure 2. A large portion of the data is sourced from existing benchmarks and synthetically corrupted. Additionally, the paper does not clarify how certain types of in-the-wild corrupted data (e.g., bright or dark illumination interference) can be systemically collected, nor does it explain why certain types of corruptions (e.g., occlusions [1]) cannot be synthetically generated.

3.	The paper states that GPT4o has a significant gap compared to human performance (Ln 502). However, according to Table 4, the differences seem relatively small (mostly within 10%) except for the MCQ task. A more detailed analysis comparing GPT-4o and human performance would provide valuable insights for the community. For example:

-- Although the paper emphasizes that LLMs perform exceptionally poorly on Step: EI, Step: CI, and Group: Wild, humans show similar patterns. How should this be interpreted? Does it imply that certain questions in the dataset become nearly impossible to answer correctly when specific types of corruption are applied? Please provide a few examples.

-- Provide detailed performance results of the MCQ task, as it shows the largest discrepancies.

Minor:

4.	The study on human robustness was conducted with only five average subjects (Ln 299), which may not be sufficiently representative.

5.	There are a few errors in the paper:

-- The compression symbols (+) are missing in Table 2.
-- Figure 3 is included but is never referenced in the paper.
-- Figure 5(b) and Figure 5(d) are identical.

[1] Ghiasi, Golnaz, et al. "Simple copy-paste is a strong data augmentation method for instance segmentation." Proceedings of the IEEE/CVF conference on computer vision and pattern recognition. 2021.

**Questions:**

1.	Please clarify the significance of the proposed relative robustness metric in response to Weaknesses-1.

2.	Please provide further details about the collected in-the-wild data in response to Weaknesses-2.

3.	Based on Weaknesses-3, please include additional discussions and details on the comparison between GPT-4o and human performance.

4.	Please double-check the manuscript in response to Weaknesses-5.

**Details Of Ethics Concerns:**

Not applicable.

---

### Note · Authors · 2024-11-27

**Comment:**

Dear Reviewers,

Thank you for your insightful comments Due to time constraints, we have not completed some of the experiments requested in the comments yet. Unfortunately, we have to **withdraw** the paper.

Anyway, we are still grateful for the inspiration provided by most of the reviewers. **We will further polish R-Bench based on the review** for a possible new submission.

Best regards,

R-Bench Author Team

**Withdrawal Confirmation:**

I have read and agree with the venue's withdrawal policy on behalf of myself and my co-authors.